# Controlling the Flight of a Drone and Its Camera for 3D Reconstruction of Large Objects

**DOI:** 10.3390/s19102333

**Published:** 2019-05-21

**Authors:** Simone Mentasti, Federico Pedersini

**Affiliations:** 1Dipartimento di Elettronica, Informazione e Bioingegneria, Politecnico di Milano, I-20133 Milano, Italy; simone.mentasti@polimi.it; 2Dipartimento di Informatica, Università degli studi di Milano, I-20133 Milano, Italy

**Keywords:** 3D reconstruction, automatic UAV flight, self-localization

## Abstract

In this paper we present a simple stand-alone system performing the autonomous acquisition of multiple pictures all around large objects, i.e., objects that are too big to be photographed from any side just with a camera held by hand. In this approach, a camera carried by a drone (an off-the-shelf quadcopter) is employed to carry out the acquisition of an image sequence representing a valid dataset for the 3D reconstruction of the captured scene. Both the drone flight and the choice of the viewpoints for shooting a picture are automatically controlled by the developed application, which runs on a tablet wirelessly connected to the drone, and controls the entire process in real time. The system and the acquisition workflow have been conceived with the aim to keep the user intervention minimal and as simple as possible, requiring no particular skill to the user. The system has been experimentally tested on several subjects of different shapes and sizes, showing the ability to follow the requested trajectory with good robustness against any flight perturbations. The collected images are provided to a scene reconstruction software, which generates a 3D model of the acquired subject. The quality of the obtained reconstructions, in terms of accuracy and richness of details, have proved the reliability and efficacy of the proposed system.

## 1. Introduction

The deployment of lightweight radio-controlled flying vehicles (UAV) is now widely spread in a large variety of application fields, like rescue or emergency operation in critical environments, professional video production, or precision agriculture. In particular, the use of UAV’s to acquire images for purposes of photogrammetry and, in general, three-dimensional reconstruction from images, represents a widely spreading application field [1,2,3,4,5]. Among the main reasons for this wide diffusion, there are the low cost and extreme versatility of modern micro-UAVs, able to fly along precise trajectories and to keep a fixed position steadily, as well as the simultaneous decreasing cost and increasing resolution of image sensors.

This paper describes the design and the implementation of a technique for autonomous mission control of a quadcopter, which carries out an autonomous acquisition of the proper image sequence for 3D reconstruction, while processing the acquired video in real time for self-localization and consequent navigation, according to the mission control. This technique has been tested on several different real objects, showing its robustness and effectiveness, as witnessed by the accuracy of the finally obtained 3D models.

### 1.1. Related Work

The research literature proposing drones for 3D reconstruction of objects and terrains considers several different sensing techniques, like high-resolution cameras [6], or a combination of cameras and laser scanners [7]. There are several state-of-the-art works [3,6,8,9,10], in which a camera is mounted on a drone which carries out its mission autonomously, exploiting the images acquired on flight both for navigation (SLAM) and for 3D reconstruction. In particular, refs. [8,9] propose quite similar applications, where there is even no need for known-shape markers previously placed in the scene. However, to achieve such powerful tasks in real time, these systems need the support of remote high-performance computing facilities, connected with the drone through a high-bandwidth, low-latency wireless link, and running remotely the SLAM and 3D reconstruction algorithms on the image stream acquired by the drone.

Conversely, the purpose of this work was to develop a *stand-alone* system for automatic image acquisition of a subject to reconstruct in 3D, where the only computing facility is the mobile device (an Android-running tablet or smartphone) normally connected to the remote controller of the drone and used as display of the drone camera. Aiming at developing a low-cost and easy-to-use system, the proposed technique has been designed to keep the workflow requested to the human operator as simple as possible. Therefore, the novelties of the proposed technique, with respect to the state-of-the-art, lie mainly in two aspects:The system has been designed to keep the workflow simple and the practical setup of the scene as less invasive as possible;The algorithm performing the UAV self-location and navigation in real time has been designed for high computational efficiency, so that it can run on the same Android device employed for UAV flight supervision.

### 1.2. Article Outline

The article is organized as follows: Section 2 discusses possible approaches and describes the proposed procedure workflow; Section 3 describes the developed algorithm; Section 4 presents some significant experimentally obtained results, and Section 5 reports some final consideration and ideas for further research.

## 2. The Acquisition Procedure

### 2.1. Scene Setup

According to the laws of multiple-view geometry [11], it is necessary to collect images of a subject from all possible viewing directions, to reconstruct its 3D shape. In general, this could be achieved by acquiring a video, or several still images (the number of images depending on the object properties and on the desired level of detail in the reconstruction), while moving around the subject, approximately following a circumference, as shown in Figure 1. For objects whose footprint is very different from a circumference (for irregular or long-and-thin footprints, for instance), a circular trajectory would be far from optimal.

The definition of the optimal fight trajectory for image acquisition, given an approximate volumetric extent of the subject, is a well-studied problem in the literature [4,10]. However, the constraint on the optimal location of the image viewpoints is not so strict; in fact, a simple and safe rule-of-thumb to obtain accurate 3D reconstructions is to maintain approximately the same distance from the local front surface of the subject.

Our aim was therefore to have the drone automatically flying at least one round along this trajectory, while (a) keeping an approximately constant distance from the floor and from the object, and (b) keeping the on-board camera always pointed toward the object. This navigation problem can be actually considered a typical “line following” problem [12], where the line to follow represents the desired closed trajectory around the object, with the horizontal projection of the camera axis kept orthogonal to the trajectory. Besides this necessary requirement, our goal in designing the acquisition procedure was to keep the practical setup of the scene as simple as possible (that is, with minimal geometric constraints on the scene). A simple solution would be to trace a line on the ground around the object, which could be tracked and followed by the drone during the acquisition procedure. However, in order to keep the operative workflow maximally simple, we propose a scene setup in which the trajectory is defined by *trajectory markers*, in form of uniform-color spheres, that are properly placed along the desired trajectory, as represented in Figure 1. The line to follow is therefore represented by the ideal line connecting adjacent trajectory markers. This choice presents some advantages, compared to tracing a line: placing spheres (e.g., common plain-color balls) on the ground around an object, with no other geometrical constraints but keeping approximately the same distance from the object itself, is significantly simpler and less invasive than tracing a line around it. Moreover, the proper choice of the marker color makes the image processing procedure for their detection and localization significantly robust against detection errors, thus improving the reliability of the whole procedure. Finally, although spherical objects are imaged as ellipses, with eccentricity raising with their distance from the optical image center [13], for the size of the imaged markers and the angular amplitude of the camera’s frustum in our case, this perspective distortion can be neglected for the sake of marker localization, assuming that the visual ray going through the centroid of the imaged ellipse coincides with the visual ray through the corresponding sphere center. This assumption is fundamental to achieve the necessary accuracy in the localization of the spherical markers, which is fundamental for the final accuracy of the reconstructed 3D model. The validity of this assumption is confirmed by the reconstruction quality of the models presented in the experimental results.

The total number of markers is arbitrary, as it depends on the particular trajectory around the subject. The size of the markers is also not critical, as it is just necessary for the markers to be clearly visible in the acquired images. Consequently, the size of the markers should be adapted to the average subject-camera distance, which in turn, depends on the size of the subject being reconstructed. In other words, it is possible to reconstruct arbitrarily large objects, provided we use correspondingly large markers, big enough to be seen from the camera. The only necessary constraint in this procedure is that at least two markers must be visible in each image, as explained in Section 3.

### 2.2. Procedure Workflow

The idea behind this work is to conceive a procedure that is as simple to carry out as possible, with minimal user intervention. The proposed image acquisition workflow consists therefore of the following steps.

#### 2.2.1. Scene Setup

The operator places a set of spheres all around the object or scene to be reconstructed. The size of the spheres should be approximately adapted to the scene’s overall size: the bigger the scene, the bigger the spheres. However their size is not critical at all: they just have to appear in the images as uniform-color circles. The color of the spheres can be arbitrarily chosen, as long as it contrasts with their background, as the marker detection procedure is adapted to the chosen color. The proper number of deployed spheres depends on the size and shape of the object, but also this choice is not critical: the spheres should be just many enough to ensure that each view sees at least two of them. A greater number of visible spheres generally increases the robustness of the procedure but does not affect the final 3D accuracy.

#### 2.2.2. Drone Lift

The user, who can control the flight seeing from the camera viewpoint in real time (by means of the control application running on the device connected to the drone), lifts the drone to the starting position, an arbitrary point of the desired trajectory. With the camera pointing towards the object, the camera should “see” the spheres lying on the ground in front of the subject, which through the perspective projection, will be normally located in the image below the object to reconstruct (as schematized in Figure 1).

#### 2.2.3. Automatic Acquisition Procedure

Once the user has placed the drone in a valid starting position, he can start the automatic acquisition procedure by giving a command to the *Mission Control* application running on the device, which takes control of the whole flight and image acquisition process. The user can monitor the ongoing process through the user interface of the application.

#### 2.2.4. Termination

During the acquisition process, the application continuously updates an estimation of the *camera attitude* (the direction of the camera axis). When, according to this estimation, the drone has completed the round along the closed trajectory, the application automatically drives the drone to land, thus completing the procedure.

## 3. The Real-Time Navigation Algorithm

### 3.1. High-Level Flight Control Strategy

Aim of the mission control algorithm is to drive the drone approximately one round along the closed trajectory, as it is defined by the markers, while: (a) keeping the object to reconstruct approximately in the image center, and (b) keeping approximately the same horizontal distance from the trajectory. Due to the constraints imposed by the drone’s flight control interface [14], the most efficient solution was to split the control into two separate correction phases. This is accomplished by the proposed algorithm, whose high-level structure is shown in the flowchart of Figure 2. The structure of the algorithm is a control loop, aimed at controlling the drone flight and the image acquisition in real time. As Figure 2 shows, the loop alternates image analysis tasks, in which the acquired photograph is analyzed by the application and all visible markers are localized, and flight control tasks, in which the application, according to the measured marker positions, issues motion commands to the drone (through the functions of the drone API [14]). A loop iteration takes approximately 6 s; on average, half of this time is taken by the image analysis tasks, and the remaining time is taken by the drone to adjust its position (when necessary) and move to the next viewpoint. Considering the average lifetime of a battery pack on the adopted drone (a *DJI Phantom 4*), the proposed procedure enables therefore to carry out acquisitions of up to 250 images with a single flight.

#### 3.1.1. Attitude Correction

After the acquisition of a camera frame, the system performs the *attitude analysis* of the acquired image, consisting of the estimation of the camera attitude from the positions of the localized trajectory markers. The details of this task are described in Section 3.1.1. If the attitude relative to the subject needs a correction, a corresponding command is given to the flight controller (a *yaw rotation*, corresponding to a rotation around the vertical axis) and the test is carried out again (see Figure 3). This cycle is repeated until the correct attitude is reached (corresponding to a centered subject in the image).

#### 3.1.2. Distance Correction

Once the drone has reached the correct attitude, a *distance analysis* is performed on the last acquired image, in order to evaluate the horizontal distance from the defined trajectory. The details of this task are described in Section 3.1.2. If the estimated distance is outside the validity range, a correction command is given, consisting of a horizontal translation of the drone along the same direction of the camera axis, in order to reach the desired distance (see Figure 4), then the distance test is repeated. Also this cycle is repeated as long as a valid distance is reached. Each of the two above correction tasks is composed of three subsequent steps:A new image is acquired and a *sphere detection* algorithm detects the visible spherical markers and localizes them accurately by estimating the centroids of the circular marker images.Using the centroid coordinates as input data, the *attitude analysis* and *distance analysis* algorithms, respectively, compute the necessary corrections.The system gives the computed attitude/distance flight correction command to the drone and waits for command execution.

These detection and analysis algorithms are now described in details.

### 3.2. Efficient Spherical Marker Localization

There is plenty of algorithms for circle detection and accurate localization in literature [15,16]. In this particular application, however, the most crucial aspect is the computational efficiency, as the algorithm runs in real time on a platform with limited computing power (a tablet running Android), so longer computation times necessarily lead to longer idle times during the drone flight. For this reason, a fast algorithm is of primary importance in this application.

Each time the drone acquires an image and the Android device receives it through the radio link, the application starts the marker detection algorithm, which is composed of the following three processing steps: *color segmentation*, *shape selection*, and *centroid localization*.

#### 3.2.1. Color Segmentation

This step is carried out by thresholding the image in the RGB color space. Different color spaces, like YUV and HSV, have been also considered for this thresholding and tested on the real images, but RGB has been finally chosen for its robustness against the high variance of the illumination, typically occurring in daylight-illuminated images. The HSV space is really not well suited for clustering constant-color regions in outdoor scenes because, rather counter-intuitively, the imaged hue of a constant color changes significantly going from sunlight to shadow, as demonstrated in [17].

Moreover, the strongly varying illumination, typical in outdoor scenes, makes it difficult to achieve a correct clustering by thresholding with an a-priori fixed threshold. For this reason, we developed an adaptive approach in which the threshold values are adapted to the current acquisition session. In order also to account for the ease-of-use design requirement, the application asks the user to take a picture of one of the deployed trajectory markers just before starting the acquisition flight. The application then extracts from this image all the pixels belonging to the well visible marker and their color coordinates are averaged, thus yielding a robust estimate of the mean marker color ${\bar{C}}_{m}=\langle {\bar{r}}_{m},{\bar{g}}_{m},{\bar{b}}_{m}\rangle$, defined as vector in the RGB space. The color segmentation criterion, applied to all subsequently acquired images, is then defined as follows: each pixel *i* of color $C_{i}=\langle r_{i},g_{i},b_{i}\rangle$ belongs to the color subspace of the markers if

(1)$$\delta_{min}\;\cdot \;{\bar{C}}_{m}\leq C_{i}\leq \delta_{max}\;\cdot \;{\bar{C}}_{m}\;.$$

The threshold factors $\delta_{min}$ and $\delta_{max}$ have been determined experimentally, aiming at the most accurate color segmentation results in daylight illumination conditions. The segmentation produces a binary image, in which the ‘white’ pixels denote pixels whose color satisfies the color similarity criterion defined in (Equation 1).

#### 3.2.2. Shape Selection

In the binary image resulting from segmentation, all white connected regions are localized using a computationally efficient region-growing algorithm. All regions then undergo a shape selection process in which a region is recognized as a valid marker if all the following criteria are satisfied:The area of the considered region $S_{i}$ (i.e., the total number of pixels) lies within the empirically measured *likelihood interval*: $S_{min}\leq S_{i}\leq S_{max}\;;$The shape of the considered region is approximately circular. The similarity to a circle is evaluated by computing its *circularity index*
$$CI_{i}=\frac{S_{i}}{{\left(\phi_{i}^{max}\right)}^{2}}\;,$$where $\phi_{i}^{max}$ denotes the maximal distance between any two points of region *i*. Considering that for a perfectly circular shape, CI would be $CI^{max}=CI\left(\mathrm{circle}\right)=\frac{\pi r^{2}}{{\left(2r\right)}^{2}}=\pi /4\;,$ a region *i* is considered valid if $CI_{i}\geq \frac{CI^{max}}{2}=\pi /8\;.$ This fixed-threshold, despite its simplicity, has proved its reliability through extensive experimental tests.

#### 3.2.3. Centroid Localization

For each region recognized in the previous step as the image of a spherical marker, the circle center is estimated by simply computing the geometrical barycenter of the region. More sophisticated estimation techniques, like the estimation of the least-squares circumference on the region perimeter, have been also tested, but they did not provided significant accuracy improvements, while requiring significantly more computation time compared to the computationally efficient barycenter estimation. Due to the prioritary processing speed requirements, we therefore stuck to the more efficient technique.

### 3.3. Estimation and Control of the Camera Attitude

The image coordinates of the detected marker centers are the input for the module computing the drone camera attitude, that is, the direction of the camera axis with respect to the current trajectory and, consequently, to the subject to reconstruct. The geometrical schematization of the problem is represented in Figure 3. If we consider the projection of the camera’s optical axis on the trajectory plane (assumed to be horizontal), the camera attitude is considered correct if the axis projection is orthogonal to the trajectory curve in their intersection point *C*, the one closer to the camera. The image of this point of intersection, in the acquired frame, corresponds to the intersection of the imaged trajectory, which can be defined as the smooth curve passing through the visible markers, with the vertical mid-line *v* passing through the image of the optical center. In fact, *v* corresponds to the intersection of the image plane with the vertical plane containing the optical axis, as the image’s horizontal axis is parallel to the horizontal plane (the camera is held horizontal by the drone active gimbal).

Under the described geometric assumptions, the local direction of the imaged trajectory in its intersection with *v* gives a direct measure of the angular deviation from the correct attitude. Referring to Figure 3, calling α the angle formed by the tangent *t* to the trajectory in the intersection point, with the image horizontal axis, α=0 (*t* is horizontal) in case of correct attitude, whereas α>0 (line *t* rising, from left to right) when the camera should be rotated to the left and, vice versa, rotated to the right for α<0 (line *t* falling).

The estimation of α would require the estimation of the interpolating curve passing through all the visible markers, whose number is significantly variable from image to image. Experiments with different interpolation models have shown that the varying number of interpolation points leads to position instabilities in case of higher-order fitting curves. For this reason, and also considering the need for fast and efficient algorithms, we adopted the simplest interpolation model, consisting of a trajectory curve defined as the piece-wise line connecting adjacent markers. According to this simplified model, α is simply and efficiently computed as the slope of the linear segment connecting the two nearest markers to the middle line *v*, on the bottom part of the trajectory, as schematized in Figure 5.

Based on the estimated value of α, the algorithm issues a yaw rotation command to the drone, in order to reach the correct attitude. For α>0, a counter-clockwise *yaw* rotation of the drone is necessary, clockwise for α<0. The amplitude of the yaw angle is computed as a function of α. An experimental search for the best function relating the determined deviation angle α to the requested yaw rotation angle ψ resulted in the observation that a simple proportional action, despite its simplicity, is one of the most effective correction strategies. Moreover, for low values of α there is no need to apply a correction, as the measured attitude allows anyway a valid image acquisition. For these reasons we adopted the following correction strategy:(2)$$\mathrm{Yawrotation}\;\;\psi =\{\begin{array}{cc} 0 & \left|\alpha \right|\leq \alpha_{min},\\ k_{P}\;\cdot \;\alpha & |\alpha |>\alpha_{min}, \end{array}$$where $\alpha_{min}$ therefore represents the threshold angle for a correcting action: an attitude correction is triggered only when |α| exceeds $\alpha_{min}$. The best value for $\alpha_{min}$ has been also determined experimentally, by searching for the value giving the minimum occurrence of corrections (and therefore the shortest overall acquisition time) without affecting the quality of the final 3D point cloud. For the adopted drone/camera setup, the best value resulted in being $\alpha_{min}=2^{\circ }$. The best value for $k_{P}$ has been obtained experimentally as well, searching for the value giving the quickest correction without incurring into oscillating behaviors in the flight dynamics due to overcorrections of the yaw angle. For this drone, the best results were obtained with $k_{P}=0.8$.

### 3.4. Estimation and Control of the Subject-Camera Distance

The correct camera attitude ensures that the subject is horizontally centered in the field of view, but gives no guarantee that the vertical extent of the subject is entirely contained in the image. A successful acquisition, for any vertical extent of the subject, is guaranteed as follows: the acquisition procedure requires the user to place the drone in an initial position, in which the vertical extent of the subject is entirely contained in the image field. After then, the automatic acquisition procedure keeps the distance and the camera pitch (the angular vertical elevation of the optical axis, corresponding to θ in Figure 4) constant, thereby ensuring that the vertical extent of the subject in the image will be kept contained in the image field.

Operatively, the constancy of the camera pitch is automatically obtained by keeping the drone in steady flight (also called *hovering*) when an image is shot. A constant distance from the subject, conversely, is achieved by keeping the vertical height of the lowest trajectory point in the image plane constant. As Figure 4 shows, if the camera pitch θ is constant, then the distance of the drone from the nearest point along the trajectory (and consequently from the subject) is $D_{T}=\frac{h}{\tan(\theta +\phi )}$ where *h* is the drone height with respect to the ground plane and ϕ is the vertical angular coordinate of the visual ray corresponding to the nearest trajectory point, $P_{T}$. Since we need a constant value for $D_{T}$, we need *h*, θ and ϕ to be constant: the procedure holds *h* constant by commanding constant height to the drone flight controller (the drone is equipped with sensors measuring the distance from ground); θ corresponds to the vertical inclination of the camera gimbal, that is held fixed, so θ is constant during hovering; ϕ is constant as long as the vertical image coordinate $y_{T}$ (see Figure 6) of the lowest trajectory point $P_{T}$ is constant.

Referring to the acquisition geometry represented in Figure 4 and Figure 6, it is possible to control the value of $y_{T}$ without changing *h* and θ, by translating the drone along the *distance control line**l* (see Figure 4), defined as the intersection between the horizontal plane and the vertical plane containing the optical axis of the camera. The distance estimation and control algorithm works therefore as follows:The algorithm selects the two centermost markers among those provided by marker detection; since this step occurs just after attitude correction, the two markers likely describe the bottommost section of the imaged trajectory. As Figure 6 shows, $y_{T}$ is estimated as the *y*-coordinate of the intersection of the segment connecting the two centermost markers with the *y* axis of the image reference frame.Depending on the amplitude of the deviation from the initial value, $\Delta y_{T}=y_{T}-y_{T0}$, the algorithm decides, as correcting action, a translation $S_{l}$ along the distance control line *l*.

Similarly as for the attitude control, the implemented correction strategy is a function relating the translation amplitude $S_{l}$ to the measured deviation $\Delta y_{T}$:$$S_{l}=\{\begin{array}{cc} 0 & |\Delta y_{T}|\leq \Delta y_{min}\\ k_{S}\;\cdot \;\Delta y_{T} & |\Delta y_{T}|>\Delta y_{min} \end{array}$$that is, a correcting action is undertaken when $\Delta y_{min}$ is exceeded. Similarly as for the attitude, the best values for $\Delta y_{min}$ and $k_{S}$ have been determined experimentally, by finding the value giving the minimum occurrence of corrections without affecting the quality of the final 3D point cloud. For the adopted drone/camera system, the best value for $\Delta y_{min}$ was 5% of the total image height, while $k_{S}\approx 2\;{\mathrm{cm}\;\mathrm{pixel}}^{-1}$.

### 3.5. Drone Flight Control

As the flowchart in Figure 2 shows, there are three different situations in which commands are issued to the drone for controlling its flight and its position: the correction of the attitude, the correction of the subject-camera distance, and the move to the next viewpoint for a shoot. To accomplish these tasks, it is necessary to access the flight control software interface, available to the application by means of the drone’s API, provided by the drone manufacturer through a Software Development Kit for mobile devices [14]. The main tool provided by the API for flight control is the function call: sendVirtualStickFlightControlData(pitch, roll, yaw, throttle, time), which simulates the corresponding actions on the controller sticks. The provided argument values (roll, pitch, yaw, and throttle) define the entity of the actions on each virtual stick, while *time* defines the duration of these actions. After that time, the actions are terminated and the drone returns to a steady hovering flight. Having such API call as interface, it is necessary to define all the motion tasks requested by our procedure, in terms of proper combination of arguments for the call.

Actually, for a given requested motion, the set of arguments leading to that motion is not unique: for instance, a change in the attitude *A* should be the result of the product A=yaw·time, as yaw defines the angular velocity around the vertical axis. The preferred solution would then be the fastest, i.e., $\langle yaw_{max}\;,\;time=A/yaw_{max}\rangle$, but we experienced that more intense actions lead to bigger uncertainties in the resulting motion. For this reason we carried out “tuning” experiments, to find the best combination of arguments for each of the motion tasks, leading to the best trade-off between motion speed and motion accuracy. Each of the three motion tasks has been therefore implemented as a call to sendVirtualStickFlightControlData, with the arguments specified in Table 1, where $roll_{OPT}$, $pitch_{OPT}$, $yaw_{OPT}$, and $thr_{OPT}$ are the optimal arguments and *A*, *D* and *L* represent the desired attitude correction, the distance correction and the lateral shift, respectively.

Concerning the desired correction parameters, *A*, *D*, and *L*, it is important to notice that while *A* and *D* result from the computation of the necessary correction, the amount of lateral motion *L* is arbitrarily set by the user, according to the desired inter-distance between subsequent image shots. The total amount of acquired images varies according to this parameter.

### 3.6. User Interface

The most important role played by the user interface of this application is to provide the user with exhaustive real-time information about the progress of the on-going acquisition procedure, thus allowing the user to supervise the whole process. A screenshot of this interface is shown in Figure 7. The interface presents a dashboard showing the video stream currently acquired by the camera and, superimposed to the image frames: (a) a vertical line representing the position of the ν axis, as shown in Figure 3 and Figure 5; (b) a horizontal line representing the line: $y=y_{T}$ in the image plane. As schematized in Figure 6, this is the line defining the target “height” of the lowest trajectory marker $P_{T}$; (c) a red segment connecting the two bottommost trajectory markers selected for attitude/distance correction; (d) the list of the latest flight commands given by the control application to the drone.

The provided information, updated in real time, allows the user to supervise the flight and to evaluate the quality of the images taken during the whole acquisition process. Through this interface, for instance, it is clearly visible how many corrections of the trajectory are needed, or whether a trajectory marker is not visible or not recognized.

## 4. Experimental Results

The purpose of the experiments and tests we carried out was to judge the obtained performance in terms of the following aspects: (a) the accuracy of the flown trajectory and of the set of acquired viewpoints, in terms of position, orientation and spacing regularity; (b) the overall duration of the acquisition process; and (c) the final accuracy of the 3D model reconstructed with the acquired images.

### 4.1. Experimental Environment

The system was developed around a *DJI Phantom 4* quadcopter; the application controls the drone through the DJI API [14]. The on-board camera of this copter is equipped with a 1/2.3${}^{\prime \prime }$ CMOS 12.4 Mpixel sensor and a wide-angle (20 mm (35 mm format equivalent), *f*/2.8 lens. The camera provides still color images with a resolution of 4000×3000 pixel. The Android device used for the experiments is a 8${}^{\prime \prime }$*Samsung Tab S4* tablet, equipped with eight *ARM Cortex-A73* cores (running at 1.9 GHz), 4 GB RAM memory and 64 GB flash memory. The tablet running our application is connected to the USB port of the drone’s remote controller, in the same way as the usual device serving as visual interface. Through this interface, the application receives the video stream and the high-resolution images from the camera, and transmits the flight commands to the drone. During the flight, the progress of the acquisition process can be followed on the application dashboard (see Figure 7).

### 4.2. Flight and Acquisition Process

The acquisition of the subjects presented as results (two sculptures, see Figure 8 and Figure 9) have been carried out in sunny daylight, in order to assess the reliability of the marker detection algorithm in presence of shadows. In the two cases, markers of two different colors have been used, to assess the flexibility of the detection algorithm. For the considered subjects, having a size of up to 4 m tall, 2 to 3 m wide, we used eight markers, placed approximately on a circumference (for the subject in Figure 8) and on an ellipse (for the subject in Figure 9).

Thanks to the achieved algorithm efficiency, the system is able to acquire and analyze one frame every 3 s, on average. For both subjects, the system collected over 200 images in total, in approximately 20 min (fast enough to complete the process within the drone battery lifetime). The overall accuracy in following the trajectory could be estimated in terms of the occurrence rate of a trajectory correction. In all the performed experiments, the acquisition process has proved to be significantly robust and accurate: a trajectory correction after the execution of the computed motion was seldom necessary.

To evaluate the ability of the system in locating the markers, we examined the video of the whole acquisition processes, available through the application dashboard. In all experiments, the detection algorithm detected all the visible markers (no “false negatives” occurred); in few situations a “false positive” occurred (mainly in backlit images) but, in all of them, the correctly detected markers kept always the trajectory estimation correct, with no significant consequence for the flight accuracy.

### 4.3. Final 3D Subject Reconstruction

Although this work focuses on the automatic acquisition process, a qualitative evaluation of the 3D reconstruction results obtained from the acquired images makes sense as a final validation of the proposed procedure. Indeed, the quality of a 3D reconstruction result is very sensitive to the quality of the acquired images and to the correct choice of their position [11]; for this reason, the result we are interested in here is the success and the overall quality of the final 3D reconstruction, rather than the quantitative measurement of the dimensional accuracy of the resulting 3D shape, which essentially depends on the characteristics of the camera and of the used 3D reconstruction software. We adopted a professional Structure-from-Motion software (*Agisoft PhotoScan* [18]) to compute these 3D reconstructions.

We selected subjects with complex shapes and irregular spatial extent, to test the procedure and the reconstruction on a challenging case. Figure 8 shows the first example. The sculpture belongs to the collection *Popolo del cibo*, by D. Ferretti, which was exposed at the international “Expo 2015” in Milan. The particular shape of this subject (3.5
m height, compared to 1.2
m width at the basement), together with the presence of fine details on the subject surface, led to a high density of the viewpoints. The drone flew a circular trajectory of approximately 4 m radius around the sculpture, acquiring over 200 views, with an average image interdistance of approx. 10–15 cm.

The images have been then processed using a professional Structure-from-Motion software [18]. The resulting quality of the reconstructed 3D models, shown in Figure 8, proves the effectiveness of the proposed technique: the 3D metric reconstruction from the acquired views was successful. The planarity of the basement faces and their mutual orthogonality have been correctly reconstructed, as well as the little details present on the complex shape of this subject.

A similarly demanding subject is the sculpture shown in Figure 9. In this case we deployed orange markers, to test the reliability of the procedure against the marker color. Due to the shape of this subject, we placed eight markers on the ground, approximately forming an ellipse around the sculpture. The drone followed therefore an elliptical trajectory, keeping the starting distance of approx. 3 m from the sculpture during the whole flight, acquiring approximately 200 views. As the 3D model in Figure 9 shows, the reconstruction process was equally successful, despite the particularly complex 3D shape of this subject.

## 5. Conclusions

This paper describes the design and implementation of a simple stand-alone system (consisting of a drone and a simple device—a tablet or a smartphone) able to acquire large-size subjects by means of a camera aboard a UAV, where both the flight and the image shooting are automatically controlled by the proposed application running on the device, which is connected to the drone through the radio link provided by the remote controller. The experimental results have shown that the system is able to accurately follow the planned trajectory, with significant robustness against outdoor illumination and flight perturbations, and to yield a sequence of images enabling good-quality 3D model reconstructions of the acquired objects. The reconstruction results for the considered complex-shaped objects is a sound proof of the reliability of this technique. Compared to the state of the art [8,9,10], the peculiarity of the proposed technique is the ability to produce a valid image data-set for 3D reconstruction with an extremely simple hardware/software system and maximally simple user operation.

Among the possible directions for further improving the proposed technique, we are considering, in particular: (a) the further optimization of the marker localization algorithm, aiming to reduce the total processing time. Since the current solution takes approximately half of the time between two subsequent shots, decreasing this time could ideally speed-up the entire acquisition process up to 2×; (b) the extension of this procedure to a “multi-level” version, able to follow multiple trajectories lying on different heights, to enable the image acquisition of particularly tall objects (like a tower, for instance).

## Figures and Tables

**Figure 1 sensors-19-02333-f001:**
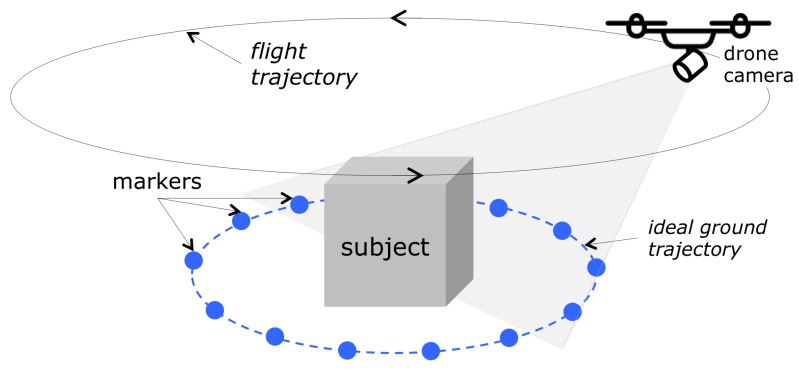
The acquisition setup. The drone has to fly at least one round along the trajectory surrounding the object to reconstruct.

**Figure 2 sensors-19-02333-f002:**
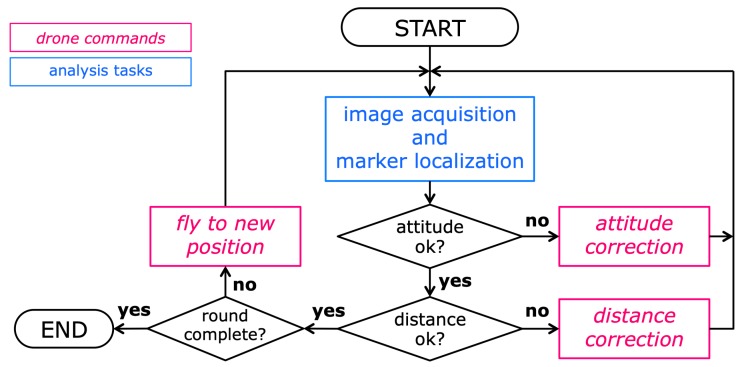
High-level flow diagram of the mission control algorithm. The blue block corresponds to image analysis tasks computed by the application in real time; the red blocks correspond to flight commands issued by the application to the drone.

**Figure 3 sensors-19-02333-f003:**
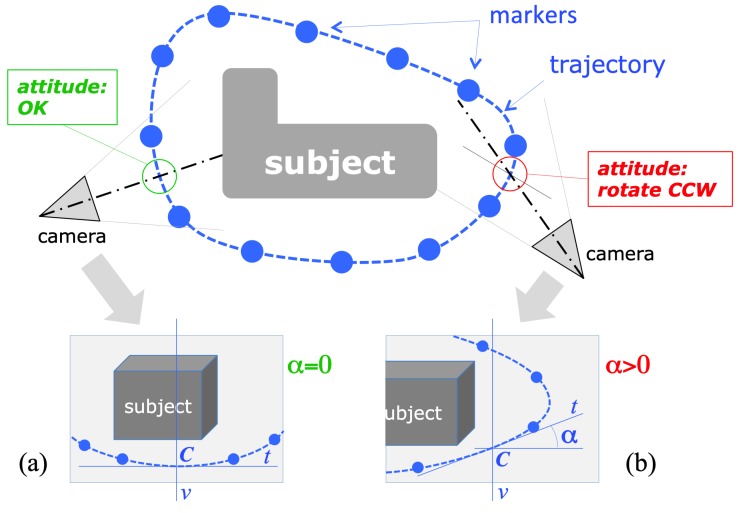
Schematization of the acquisition geometry, for the estimation of the actual camera attitude. The angle α (defined in the images as the angle between the tangent to the trajectory in *C* and the horizontal) is a measure of the deviation from the correct attitude: (**a**) α=0: correct attitude; (**b**) α>0: drone should rotate to the left (counter-clockwise yaw rotation).

**Figure 4 sensors-19-02333-f004:**
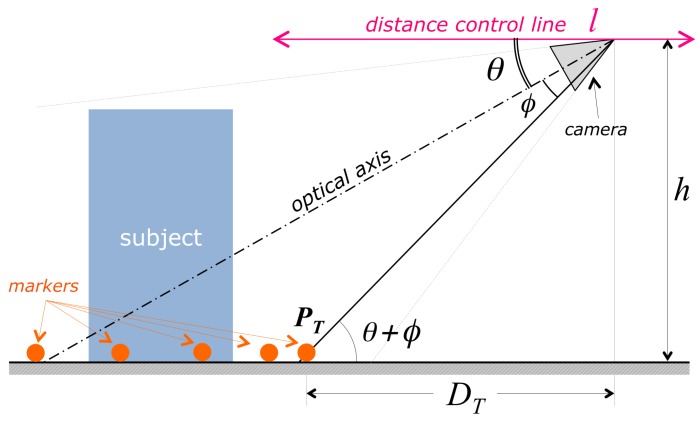
Schematization of the acquisition geometry, regarding the estimation and control of the subject-camera distance. Aim of the flight control procedure is to keep the horizontal camera-trajectory distance $D_{T}$ constant.

**Figure 5 sensors-19-02333-f005:**
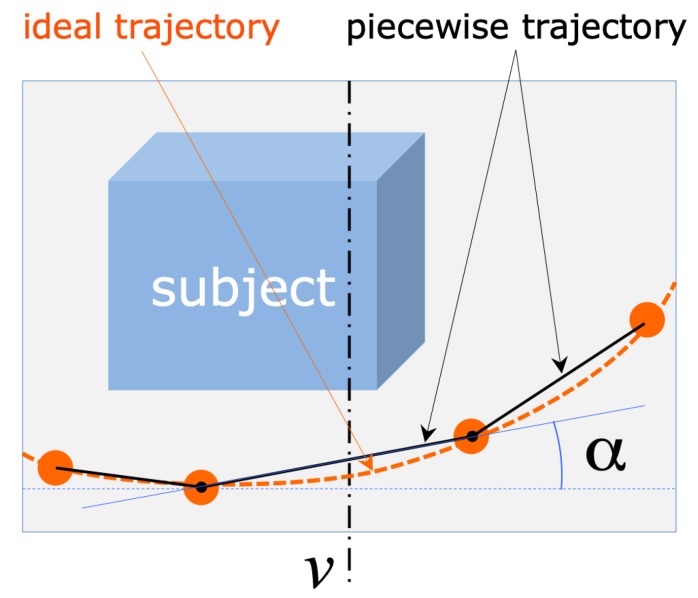
Efficient estimation of the angle α from the position of the trajectory markers: α is simply the slope of the linear segment connecting the two markers lying nearest and on opposite sides of *v*.

**Figure 6 sensors-19-02333-f006:**
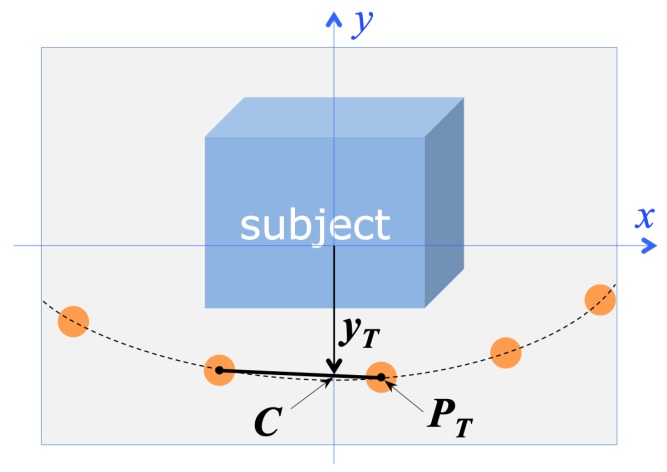
Acquisition geometry: estimation of the horizontal camera-trajectory distance $D_{T}$, from the analysis of the camera view.

**Figure 7 sensors-19-02333-f007:**
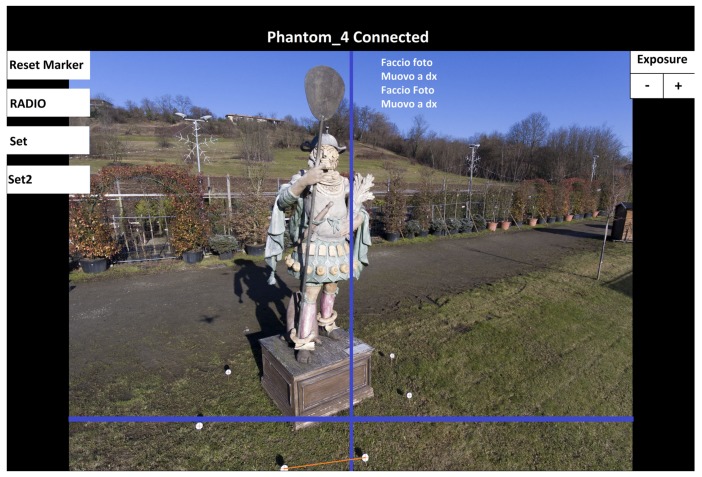
A screenshot of the dashboard of our application, during an acquisition procedure. The application dashboard shows the status of the acquisition process, like the localized markers (colored in white), the geometrical parameters (see Figure 5 and Figure 6), and the drone command history.

**Figure 8 sensors-19-02333-f008:**
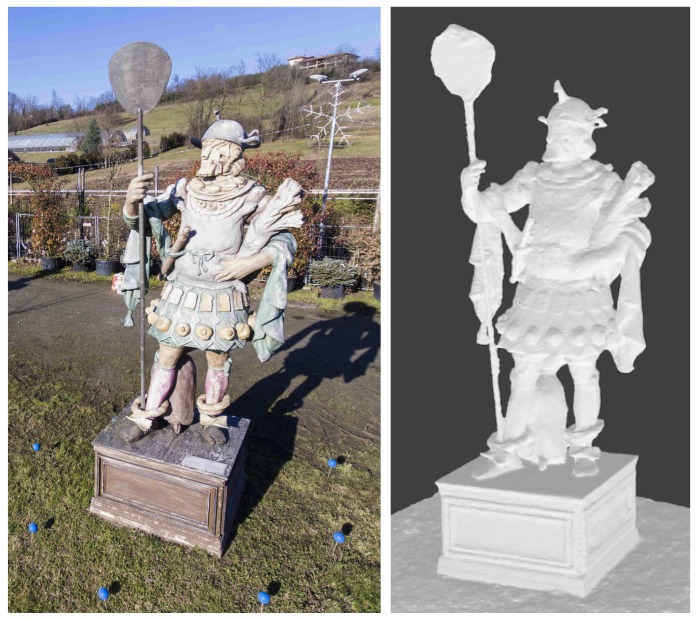
(**Left**) one of the acquired images from a sculpture by D. Ferretti (Expo 2015, Milan, Italy). Due to its complex shape, this represented a particularly difficult subject to reconstruct with good accuracy. (**Right**) a view of the reconstructed 3D model of the sculpture.

**Figure 9 sensors-19-02333-f009:**
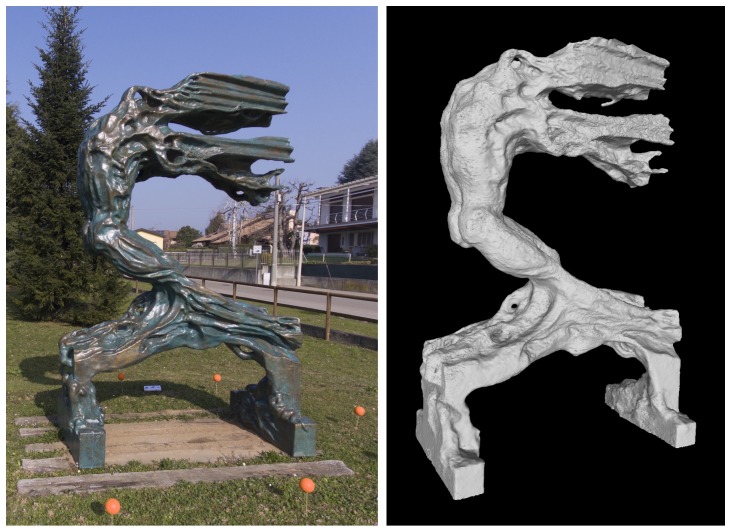
(**Left**) one of the acquired images from another sculpture (from the collection: *warriors in the wind*, by S. Volpe, Milan, Italy). (**Right**) A view of the reconstructed 3D model.

**Table 1 sensors-19-02333-t001:** Arguments passed to the sendVirtualStickFlightControlData() call, for each motion command.

Motion	Roll	Pitch	Yaw	Throttle	Time
**Attitude correction:**	0	0	$yaw_{OPT}$	0	$\frac{A}{yaw_{OPT}}$
**Distance correction:**	0	$pitch_{OPT}$	0	$thr_{OPT}$	$\frac{D}{pitch_{OPT}\;\cdot \;thr_{OPT}}$
**Lateral shift to next position:**	$roll_{OPT}$	0	0	$thr_{OPT}$	$\frac{L}{roll_{OPT}\;\cdot \;thr_{OPT}}$

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
