# Peer review of "Controlling the Flight of a Drone and Its Camera for 3D Reconstruction of Large Objects"

_sensors, 2019, doi:10.3390/s19102333_

Reviewer 1 Report

Thanks to the author for the article. Your paper presents an interesting proof of concept of a method to scan a large object and presents successful results on experiments. However, for the journal, it would be good to improve:

- section 3. Give more detail on the algorithm. The section gives a good high level (with some low level details) but it lacks a more algorithmic description of your proposal. In particular, how do you compute height and positions to stop, are those given by the 3D scanning software or do you chose a fixed number of positions around the target? your algorithm is not stated in terms on how you feed the DJI controller with the actual positions to move the dron there. 

- A more scientific view of the algorithms and results should be discussed in section 4: how good is the algorithm? what are the limitations? it seems you detect a fixed number of balls, but what about if there are occlusions of them or ambiguities in their position? 

- Is there any parameter (ie number of balls, other variables or thresholds you set) that determine the quality of the 3D recovered? 

There are other previous approaches in the literature to scan 3D objects (without a dron), that should be mentioned in the background, and probably some additional experiments and discussions that could be considered in your article. 

Regards

Author Response

Answers to Reviewers

General remark:  
In order to evidentiate the changes in the revised manuscript, the modified text have been highlighted as
blue text.

Answers to Reviewer #1

Thanks to the author for the article. Your paper presents an interesting proof of concept of a method to scan a large object and presents successful results on experiments. However, for the journal, it would be good to improve:

- section 3. Give more detail on the algorithm. The section gives a good high level (with some low level details) but it lacks a more algorithmic description of your proposal. In particular, how do you compute height and positions to stop, are those given by the 3D scanning software or do you chose a fixed number of positions around the target? your algorithm is not stated in terms on how you feed the DJI controller with the actual positions to move the dron there.

The height of the flight is decided by the user, who sets the starting position of the drone. Afterwards, during the acquisition, the drone flight controller hold constant height from the floor, as explained in Section 3.4. The procedure allows to set a horizontal displacement between two subsequent shots. The total number of positions is not fixed.

To better clarify these points, the description of the algorithm has been extended in several point of Section 3. In particular, a subsection has been added with the details about drone control (Section 3.5 – Drone Commands).

- A more scientific view of the algorithms and results should be discussed in section 4: how good is the algorithm? what are the limitations? it seems you detect a fixed number of balls, but what about if there are occlusions of them or ambiguities in their position?

Section 4 has been restructured and significantly expanded, giving also answers to the above questions. In particular, the number of balls is arbitrary and depends on extent and shape of the desired trajectory around the subject.

- Is there any parameter (ie number of balls, other variables or thresholds you set) that determine the quality of the 3D recovered?

All the parameters (i.e. constants, thresholds) have been tuned to maximize the robustness of the automatic flight and acquisition process. These choices, however, have no direct influence of the 3D reconstruction quality, which essentially depends on the photographic quality of the images (camera resolution, image sharpness, correct exposition, illumination, etc.) and on the particular reconstruction software (in our case, reference [18]). This fact has been better explained in Section 4.3.

There are other previous approaches in the literature to scan 3D objects (without a dron), that should be mentioned in the background, and probably some additional experiments and discussions that could be considered in your article.

There is indeed a huge literature on 3D reconstruction from views, but the focus of this paper is not on 3D reconstruction, for which we used a state-of-the-art commercial software (see Reference [16]), but on the development of a simple system (with simple hardware) and a simple technique for automatic drone control and acquisition of the necessary images. We have now explicitly elaborated on this point, with a dedicated subsection “Related work” (Section 1.1).

Reviewer 2 Report

The paper describes a system to capture images of a subject aiming 3D reconstruction. The system controls a drone´s flight and the cameras viewpoint, while processing the acquired video in real time for self-localization and consequent. First, a brief introduction and related works are presented.  Next, a description of the acquisition procedure is presented, followed by a description of the navigation algorithm, and experimental results. Finally, some conclusions are discussed.

The overall structure of the paper is fine. Title is appropriate and matches the content of the paper. Even though some corrections need to be addressed throughout the paper, the English is acceptable, and the ideas are clearly transmitted to the reader. References are recent, relevant and appropriate in number.

•    Figures are relevant, but sometimes they appear far from the text describing it (i.e. Fig 3 and 5 are mentioned in page 4, but they only presented in pages 7 and 8).
•    Figures numbering should be sequential. In the text, Fig. 5 is mentioned before Fig. 4.

The proposed system presents clear practical application, but it´s the scientific contribution is not clear in the text. General remarks:

•    Although authors cite related works in the introduction, it limited to one line. The contributions of the manuscript should be described based on a proper analysis of the state-of-the-art.
•    The Conclusions section is rather brief and thus limited. In this section one would expect to find at least a deeper analysis of the results and future work directions.
•    The title express that the methodology is designed for the reconstruction of large 3D objects. The term large, though, is never defined in the text. What is large for the authors? a table? a house? a dam? It is important to describe it clearly so one can understand the limitations of the study. Does it work for small objects? Why it would be different to reconstruct smaller objects?
•    What is the maximum distance markers are detected? Wouldn´t it limit the use of the proposed methodology in really large objects (i.e. dams)?
•    Why choosing visual markers instead of GPS coordinates to define the UAV path? Although it is not explicit in the text, large objects are generally located outdoors, and thus GPS is probably available.
•    Depending on the object image acquisition at one fixed height might not be enough to reconstruct all details of the subjects, which is also the case for tall objects. How does the proposed methodology tackle such scenarios?
•    How the number of images collected is defined? How the distance from the subject is defined?
•    Title of section 3 uses claim that the proposed navigation algorithm works in “real-time”. Real-time constraints are very hard to achieve, and the term is never explored in the text. How does the system work in real-time when each camera frame takes 3 seconds (page 9, section 4.1) to be analyzed?
•    Several times authors claim that the algorithms run in limited computing power, but no details about the processing unit are provided (Processor, memory, OS, etc).
•    How the lower level control is performed?
•    For the sake of repeatability and proper understanding the experiments should be described in more detail. What specs of the camera? What was the conditions of the scene during the acquisition?
•    Authors claim that the resulting 3D models are accurate, but only a qualitative analysis is performed.

Author Response

Answers to Reviewers

General remark:  
In order to evidentiate the changes in the revised manuscript, the modified text have been highlighted as
blue text.

Reviewer #2

The paper describes a system to capture images of a subject aiming 3D reconstruction. The system controls a drone´s flight and the cameras viewpoint, while processing the acquired video in real time for self-localization and consequent. First, a brief introduction and related works are presented.  Next, a description of the acquisition procedure is presented, followed by a description of the navigation algorithm, and experimental results. Finally, some conclusions are discussed.

The overall structure of the paper is fine. Title is appropriate and matches the content of the paper. Even though some corrections need to be addressed throughout the paper, the English is acceptable, and the ideas are clearly transmitted to the reader. References are recent, relevant and appropriate in number.

•    Figures are relevant, but sometimes they appear far from the text describing it (i.e. Fig 3 and 5 are mentioned in page 4, but they only presented in pages 7 and 8).

Figures numbering should be sequential. In the text, Fig. 5 is mentioned before Fig. 4.

The text has been reformatted and the images are now correctly placed and in sequential order. Figures 3–6 are mentioned several times in different pages, so for these Figs. it’s not possible to get both figures and their references always in the same page (lastly, LaTeX has the last word on where the figures have to be…)

The proposed system presents clear practical application, but it´s the scientific contribution is not clear in the text. General remarks:

•    Although authors cite related works in the introduction, it limited to one line. The contributions of the manuscript should be described based on a proper analysis of the state-of-the-art.

A more detailed discussion on related work in literature has been added to the text, on a dedicate subsection (Section 1.1).

•    The Conclusions section is rather brief and thus limited. In this section one would expect to find at least a deeper analysis of the results and future work directions.

The Conclusions have been extended, discussing the obtained results and possible directions for further research.

•    The title express that the methodology is designed for the reconstruction of large 3D objects. The term large, though, is never defined in the text. What is large for the authors? a table? a house? a dam? It is important to describe it clearly so one can understand the limitations of the study. Does it work for small objects? Why it would be different to reconstruct smaller objects?

•    What is the maximum distance markers are detected? Wouldn´t it limit the use of the proposed methodology in really large objects (i.e. dams)?

With “large objects”, we just meant objects that are too big to be photographed from each side (a necessary condition for successful 3D reconstruction) simply using a camera held by hand. They normally need to be shot from higher viewpoints, that could be reached, for instance, by a drone carrying a camera. This has been now explained at the very beginning of the paper, in the introductory part of the abstract.

Apart from these aspects related to the image acquisition process, the proposed method is substantially size-independent, as long as the markers are clearly visible. In other words, it could be possible to reconstruct a dam, provided that it is surrounded by markers that are big enough to be visible in the acquired images. Conversely, considering ‘little’ objects, in this case it would be no need to use a drone for shooting the images, therefore the technique proposed in this paper would be simply unnecessary.

Specific considerations about the size of the object to reconstruct has been now added in the paper (end of Section 2.1).

•    Why choosing visual markers instead of GPS coordinates to define the UAV path? Although it is not explicit in the text, large objects are generally located outdoors, and thus GPS is probably available.

The instantaneous GPS position provided by standard receivers (like those available on common off-the-shelf drones) is affected by errors with a standard deviation of several meters (we made some experiments to measure this error), which is not accurate enough to follow the desired trajectory. But even if the GPS position were more accurate, the relevant information for our approach is the viewpoint position and orientation with respect to the placed markers, whose GPS coordinate is unknown. A GPS-based approach would then need to know the GPS position of all markers: this would make such an approach much cumbersome than the one we propose and, however, it would provide just the position and no orientation.

•    Depending on the object image acquisition at one fixed height might not be enough to reconstruct all details of the subjects, which is also the case for tall objects. How does the proposed methodology tackle such scenarios?

This is actually a limitation of our approach: the trajectory is planned to lie on the horizontal plane on which the drone is initially positioned by the user, that is, the starting position of the automatic procedure. Although the very satisfying results obtained on the considered complex subjects, it would be very interesting to make this approach capable of ‘multilevel’ trajectories. As we have now specified in the Conclusions, this is one of our primary further research goals.

•    How the number of images collected is defined? How the distance from the subject is defined?

The number of collected images is not a-priori defined. It depends on the trajectory, defined by the user through the positioning of the markers. As explained in Section 3, the procedure regulates the flight in order to keep the vertical image location of the visible markers rather constant. This, together with the constant drone height, keeps the camera always at the same approximate distance from the trajectory (as it is defined by the markers) and thus from the subject. We added some text in Section 3 to better clarify this point.

•    Title of section 3 uses claim that the proposed navigation algorithm works in “real-time”. Real-time constraints are very hard to achieve, and the term is never explored in the text. How does the system work in real-time when each camera frame takes 3 seconds (page 9, section 4.1) to be analyzed?

We have now better clarified this point in Section 3.1. Three seconds is the maximal duration of the image analysis and marker localization tasks; approximately the same time is taken by the drone command tasks, so that, on average, the system takes 5 to 6 seconds for each shot.

•    Several times authors claim that the algorithms run in limited computing power, but no details about the processing unit are provided (Processor, memory, OS, etc).

The reviewer is right. The requested details have been now added to the text, in a dedicated subsection (Section 4.1)

•    How the lower level control is performed?

We are not sure to understand the meaning of “lower level control”: we guess, the reviewer means a lower level of the battery. In this case, the procedure interrupts the acquisition if a low battery signal is received. The procedure, however, was conceived to be fast enough to complete the acquisition process within the autonomy of a charged battery pack.

•    For the sake of repeatability and proper understanding the experiments should be described in more detail. What specs of the camera? What was the conditions of the scene during the acquisition?

These details have been now added in the paper (Section 4.1).

•    Authors claim that the resulting 3D models are accurate, but only a qualitative analysis is performed.

Indeed we state in the paper that the resulting 3D quality is accurate just from a qualitative point of view, because the focus of the article is on the automatic procedure for drone flight control. The 3D reconstruction process is not a part of the novel work.

A quantitative accuracy assessment is out of the scope of this paper, as the focus of this work is the automatic procedure for acquiring proper images for 3D reconstruction purposes.

From a quantitative point of view, the 3D reconstruction quality depends essentially on the photographic quality of the images (camera resolution, image sharpness, correct exposition, illumination, etc.) and on the particular reconstruction software (in our case, reference [18]).

This point has been now discussed more explicitly in Section 4.3.

Reviewer 3 Report

In this paper, the authors propose "Controlling the Flight of a Drone and its Camera

for 3D Reconstruction of Large Objects." The collected images are provided to a scene reconstruction software, which generates a 3D model of the acquired subject. Both the drone flight and the choice of the viewpoints for shooting a picture are automatically controlled by the developed application, running on an Android device wirelessly connected to the drone and controlling the entire process. However, there are some issues should be addressed.

1. Why does this system adopt HSV color space that is suitable for color segmentation?

2. How to solve real time processing problem in this system?

3. The paper should be compared with the related works to make sure the quality of the proposed scheme.

4. The reference should follow the same format.

Author Response

Answers to Reviewers

General remark:  
In order to evidentiate the changes in the revised manuscript, the modified text have been highlighted as
blue text.

Reviewer #3

In this paper, the authors propose "Controlling the Flight of a Drone and its Camera

for 3D Reconstruction of Large Objects." The collected images are provided to a scene reconstruction software, which generates a 3D model of the acquired subject. Both the drone flight and the choice of the viewpoints for shooting a picture are automatically controlled by the developed application, running on an Android device wirelessly connected to the drone and controlling the entire process. However, there are some issues should be addressed.

1. Why does this system adopt HSV color space that is suitable for color segmentation?

We carried out experiments with several color spaces and finally obtained the best results in the RGB color space. Actually, HSV was also our first choice, as the algorithm needs to find regions with uniform hue, but we found out experimentally that clustering in the HSV space yields poorer results, probably because the Hue channel in th HSV-converted image is much noisier than the native RGB channels. This point has been discussed in the paper and now further expanded, at the beginning of Section 3.2.1.

2. How to solve real time processing problem in this system?

We have now better clarified this point in Section 3.1: thanks to the developed fast algorithms for marker localization, the system needs approximately three seconds to localize the markers and, in total, approx. 6 seconds for a loop iteration, that is, 6 seconds between two subsequent shots.

3. The paper should be compared with the related works to make sure the quality of the proposed scheme.

We have now further elaborated on state-of-the-art in this field, adding a dedicated subsection on Related Work (Section 1.1).

4. The reference should follow the same format.

We are not sure we understood this point correctly. Does the reviewer mean that all the references do not follow the same format? If yes, which references are meant? We have no problem in correcting that, if we have indications about where the problem is.

Round  2

Reviewer 1 Report

Good improvements to the manuscript, I have no further comments. 

Author Response

We would like to thank the Reviewer for the revision, whose comments were very helpful and have led to a significant improvement of our manuscript.

Thank you so much,

Federico Pedersini

Simone Mentasti

Reviewer 2 Report

The current version of the manuscript has been much improved. I have no further considerations to add.

Reviewer 3 Report

The authors do not solve the related problems that I suggest. Besides, the paper does not be compared with the related works to make sure the quality of the proposed scheme. The paper is not complete.

Author Response

Revision 2

Answers to Reviewer

Comments of Reviewer 3 (Revision 2):

The authors do not solve the related problems that I suggest. 

Besides, the paper does not be compared with the related works to make sure the quality of the proposed scheme. The paper is not complete.

Answers to Reviewer 3 (Revision 2)

Point 1: The authors do not solve the related problems that I suggest. 

Response 1: We cannot exactly understand which “related problems that I suggest” are meant by the Reviewer, that have not been solved by us. 

In Revision 1 (attached below, for completeness), we actually addressed the first three of the four points raised by the Reviewer, concerning, respectively, 1) the use of RGB space instead of HSV,  2) real-time processing problems, and 3) related work. For each of them, we gave a response and extended correspondingly the discussion on the related parts of the manuscript, to the best of our knowledge. Therefore, if the ‘related problems’ deal with one of these three points, we would like to ask the Reviewer to tell us more precisely and concretely what he expects us to be done, to solve them. 

Conversely, if the problems concern Point 4 (the ‘same reference format’), as this point was already unclear to us in Revision 1 (therefore we asked in Revision 1 for a more precise description of the problem) we would like to ask the Reviewer again to give us just more detailed information on which format problems he means, to allow us to fix it.

Point 2: Besides, the paper does not be compared with the related works to make sure the quality of the proposed scheme. The paper is not complete.

Response 2: We have now further modified Section 1.1 (Related Work), explaining more explicitly the differences between our approach and those considered the state-of-the-art in literature.

To the best of our knowledge, apart from the research work now reported in Section 1.1 there are no other significant approaches that can be meaningfully compared to our system, as we did not find in literature any system performing, like ours, real-time navigation as a stand-alonesystem, that is, without the support of any remotely-connected high-performance computational facilities processing the received video stream in real time, like all state-of-the-art approaches do.

Moreover, even when considering interesting to compare those approaches to ours (despite their substantial difference), it would be impossible to organize any comparative experimental campaign, as those systems are not available to the research community. 

We cannot therefore see other significant ways to further improve the discussion on Related Work in the manuscript. If the Reviewer still sees other aspects on which the Related Works could be significantly improved, we would like to ask him to provide us a concrete description of what kind of work still has to be done in his opinion.

==============================================================

Attachment: Revision 1

============================================================== 

Answers to Reviewer 3 

In this paper, the authors propose "Controlling the Flight of a Drone and its Camera for 3D Reconstruction of Large Objects." The collected images are provided to a scene reconstruction software, which generates a 3D model of the acquired subject. Both the drone flight and the choice of the viewpoints for shooting a picture are automatically controlled by the developed application, running on an Android device wirelessly connected to the drone and controlling the entire process. However, there are some issues should be addressed.

1. Why does this system adopt HSV color space that is suitable for color segmentation?

We carried out experiments with several color spaces and finally obtained the best results in the RGB color space. Actually, HSV was also our first choice, as the algorithm needs to find regions with uniform hue, but we found out experimentally that clustering in the HSV space yields poorer results, probably because the Hue channel in th HSV-converted image is much noisier than the native RGB channels. This point has been discussed in the paper and now further expanded, at the beginning of Section 3.2.1.

2. How to solve real time processing problem in this system?

We have now better clarified this point in Section 3.1: thanks to the developed fast algorithms for marker localization, the system needs approximately three seconds to localize the markers and, in total, approx. 6 seconds for a loop iteration, that is, 6 seconds between two subsequent shots.

3.The paper should be compared with the related works to make sure the quality of the proposed scheme.

We have now further elaborated on state-of-the-art in this field, adding a dedicated subsection on Related Work (Section 1.1).

4.The reference should follow the same format.

We are not sure we understood this point correctly. Does the reviewer mean that all the references do not follow the same format? If yes, which references are meant? We have no problem in correcting that, if we have indications about where the problem is.

Round  3

Reviewer 3 Report

The authors have solved the related problems.